# Diamond-Based Electrodes for Detection of Metal Ions and Anions

**DOI:** 10.3390/nano12010064

**Published:** 2021-12-27

**Authors:** Muthaiah Shellaiah, Kien Wen Sun

**Affiliations:** Department of Applied Chemistry, National Yang Ming Chiao Tung University, Hsinchu 300, Taiwan; muthaiah1981@nctu.edu.tw

**Keywords:** electrochemical assay, BDDE, metal ions detection, anions quantification, nanofabrication, real analysis, boron doped electrodes, sp^2^-carbon insertion

## Abstract

Diamond electrodes have long been a well-known candidate in electrochemical analyte detection. Nano- and micro-level modifications on the diamond electrodes can lead to diverse analytical applications. Doping of crystalline diamond allows the fabrication of suitable electrodes towards specific analyte monitoring. In particular, boron-doped diamond (BDD) electrodes have been reported for metal ions, anions, biomolecules, drugs, beverage hazards, pesticides, organic molecules, dyes, growth stimulant, etc., with exceptional performance in discriminations. Therefore, numerous reviews on the diamond electrode-based sensory utilities towards the specified analyte quantifications were published by many researchers. However, reviews on the nanodiamond-based electrodes for metal ions and anions are still not readily available nowadays. To advance the development of diamond electrodes towards the detection of diverse metal ions and anions, it is essential to provide clear and focused information on the diamond electrode synthesis, structure, and electrical properties. This review provides indispensable information on the diamond-based electrodes towards the determination of metal ions and anions.

## 1. Introduction

Detection and quantification of hazardous pollutants, biomolecules, drugs, herbicides, metal ions, and anions are essential to maintaining environmental sustainability [1,2,3,4,5,6,7,8,9,10,11,12,13]. A number of methods involving organic nanoprobes, covalent–organic frameworks (COFs), metal–organic frameworks (MOFs), metal nanoparticles, hybrid nanomaterials, small molecules, and polymers were engaged in the quantitation of specific analytes [6,7,8,9,10,11,12,13]. On the other end of the spectrum, instrumental tactics, such as inductively coupled plasma mass spectrometry, high-performance liquid chromatography, gas chromatography, atomic absorption spectrometry, electrochemical studies, and immunoassays have been pronounced as the conventional cost-effective approaches [14,15,16,17,18,19,20]. Among them, electrochemical-based detection of specified analyte detection seems to be impressive in terms of its selectivity and sensitivity with lower detection/quantification limits [21,22,23]. Moreover, the majority of the electrochemical assays were attributed to the electrodes employed [24,25,26]. Among those exceptional electrodes, diamond-based electrodes are noteworthy due to their remarkable performances in many analytical studies [27,28]. Moreover, nanodiamond (ND)-based materials and electrodes also possess some unique properties and utilities, as described next.

Diamond, as an unique material with sp^3^ hybridization, consists of a tetrahedrally connected carbon atomic network and exists in diverse nanostructural forms, such as nanoparticles, graphitized hybrid nanoflakes, nanocrystals, nanowires, etc. [29,30,31,32,33]. Nanodiamonds with nitrogen vacancy center (NV^−^) or surface modification can be employed in drug delivery, cell tracking/imaging, and sensing studies [34,35,36,37,38]. Similarly, diamond nanoparticle (DNP)-conjugated hybrid materials have also been reported for sensor applications [39,40]. On the other hand, diamond nanomaterials have also been applied for semiconductor applications [41,42,43,44]. For example, self-assembly of surface-modified DNPs led to diamond nanowires (DNWs) formation with distinct transport properties [45,46]. Apart from earlier mentioned applications, nanodiamond-based electrodes were recognized as an exceptional finding with unique electrochemical utilities [47,48,49,50]. In particular, the boron-doped diamond (BDD) electrodes were noted as low-biofouling materials and were engaged in the electrochemical quantification of metal ions, anions, biomolecules, drugs, environmental hazards, pesticides, organic molecules, etc. [27,28,51,52,53,54,55,56,57,58,59,60].

As for the analytical importance of diamond-based electrodes, although many specific reviews are available to describe their analyte detection ranges and detection/quantification limits [51,52,53,54,55,56,57,58,59,60], the majority of the reviews did not discuss and provide clear information on the diamond-based electrode-facilitated detection of metal ions and anions. Therefore, this review discloses details of electrochemical discrimination of metal ions and anions by diamond-based electrodes, as illustrated in Figure 1.

## 2. Diamond-Based Electrodes in Metal Ions Detection

Due to the high polluting effect of metal ions, diamond-based electrodes have been engaged in many heavy metal ions detection and quantification studies, as described in this section. Two decades ago, Ponnuswamy et al. described the consumption of diamond as a quasireference in the electrochemical sensing studies towards heavy metal ions [61], wherein polycrystalline diamond film was firstly prepared on a p-type Si substrate (with (100) crystal orientation; 0.01 Ω cm sheet resistance). Before electrochemical studies, the substrate was treated with 10% HNO_3_ for 10 min and etched in 4.9% HF for 5 min followed by washing with water. By using diamond as a quasireference electrode in 0.01% HF, the silicon sensor responded to Ag^+^ from 91 to 910 pM (pM = picomole; 10^−12^ M). Although this work is impressive, interrogations, such as interference, sensitivity, and real-time applicability still require more attentions. Thereafter, boron-doped diamond/nanodiamond (BDD) electrodes were noted as exciting materials towards Ag^+^ quantification. For example, Maldonado and coworkers compared the Ag^+^ sensing effect of planar BDD and disk BDD electrodes via differential pulse anodic stripping voltammetry (DPASV) interrogations [62]. Planar boron-doped nanocrystalline diamond film (3–4 µm thickness) was deposited over the B-doped p-type Si (100; 0.001 Ω cm) substrate via the microwave-assisted chemical vapor deposition (CVD) technique, wherein 10 ppm B-atoms were doped using 0.1% B_2_H_2_ diluted in H_2_ (at a deposition pressure = 35 Torr; micropower of 800 W; flow rates of 2.00 sccm CH_4_, 2.00 sccm B_2_H_6_ diluted in H_2_ (0.1%), and 196 sccm H_2_; sccm = standard cubic centimeters per minute). On the other hand, the disk BDD electrode was obtained from the MSU Fraunhofer Center for Coatings and Diamond Technologies. Both planar and disk BDD electrodes were employed in the detection of Ag^+^ (by DPASV; scan range = 0 to 0.6 V at 40 mV/s scan rate; Ag/AgCl (3M KCl)—reference electrode; graphite rod—counter electrode) which displayed linear ranges between 45.5 Nm–2.3 µM and 9.1–682 nM (µM = micromole; nM = nanomole) and LODs of 31 and 43 nM, respectively, at deposition time of 120 s. Figure 2 shows the DPASV responses of planar/disk BDD at diverse concentrations of Ag^+^ in in 0.1 mol L^−1^ acetate buffer (pH 4.6). Judging from the results, the disk BDD electrode seems to be more impressive in terms of its easy fabrication, sensitivity (10.6 ± 0.5 nA L µg^−1^ (planar BDD) and 3.6 ± 0.7 nA L µg^−1^ (disk BDD)), and real-time applicability with certified National Institute of Standards and Technology (NIST) solution and National Aeronautics and Space Administration (NASA) water samples.

Recently, a 2000 ppm boron-doped diamond electrode with 0.75 × 10^–3^ Ω m resistivity was demonstrated for Ag^+^ ions detection by means of DPASV response [63]. In this report, Ag/AgCl (3M KCl) and platinum were used as reference and counter electrodes, correspondingly. In the presence of 0.1 M HNO_3_, the highest DPASV signal was observed and feasible detection of silver ions in 0.1 M Na_2_S_2_O_3_ via [Ag(S_2_O_3_)_2_]^3−^ complex formation was proposed. The work displayed a linear response of 1–7 nM with an LOD of 0.2 nM (for 240 s deposition time; deposition potential = −0.18 V; scan range = −0.18 to 0.55 V at a 10 mV s^−1^ scan rate). Moreover, the suitability of electrode was also demonstrated by spiked real-sample analysis and possible interference studies. Anodic stripping voltammetry (ASV)-based detection of Ag^+^/Cu^2+^ and Ag^+^/Pb^2+^ was proposed by using diamond/graphite film and BDD, respectively [64,65]. The unique hybrid diamond/graphite nanostructured film electrode displayed a significant ASV response to Ag^+^ and Cu^2+^ [64], wherein the deposition potentials of Ag^+^ and Cu^2+^ were −0.1 and −0.4 V, respectively, with a 3 min deposition time (scan window: −0.2–0.4 V at 20 mV s^−1^ scan rate; Pt-wire—counter electrode; Ag/AgCl (3 M KCl) and Ag/Ag+ (0.01 M)—reference electrode in 1 mM [Fe(CN)6]^3−^/^4−^ in 0.1 M KCl and 1 mM ferrocene in 0.1 M tetra-butylammonium tetrafluoroborate (TBABF_4_) in CH_3_CN solution). Note that the sp^2^/sp^3^ ratio of hybrid diamond/graphite film plays a vital role in the detection of Ag^+^ and Cu^2+^, which has been demonstrated by Raman spectroscopic studies. Figure 3 shows the linear anodic stripping voltammetry (ASV) response of Ag^+^ and Cu^2+^ between 0.91 pM–9 nM and 0.16 pM–15.7 nM with LODs of 0.52 and 0.88 pM, correspondingly. Although this work was demonstrated in real tap-water sample analysis, the interference studies were incomplete. On the other hand, information of morphology, optimization, doping concentration, and fabrication details on the BDD electrode-mediated detection of Ag^+^ and Pb^2+^ [65] are still insufficient, thereby requiring more attention in validating its efficacy.

Detection and quantification of As^3+^ were demonstrated using iridium-implanted/modified BDD and Au/AuNPs-coated BDD electrodes [66,67,68,69,70]. Ivandini et al. applied the iridium-implanted BDD (Ir-BDD; B/ C = 1:100, implanted with 800-keV Ir^+^ with a dose of 10^15^ cm^−2^) electrode for effective detection of As^3+^ via amperometric and flow injection analysis [66]. The Ir-BDD electrode showed catalytic activity to As^3+^ (potential window = −0.8 to + 1.0 V; Ag/AgCl/1 M LiCl as a reference electrode in 0.1 M phosphate buffer) with a linear response from 0.1–100 µM and an LOD of 20 nM. This work was demonstrated by tap water interrogations, but the interference studies were not conducted. Gold-coated BDD (Au-BDD; B atom doped at a concentration of 10^20^ cm^−3^ with resistivity of 0.01 Ω cm) electrode was utilized in determination of As^3+^ ions via DPASV responses [67]. Herein, Ag/AgCl and graphite rods acted as reference and counter electrodes, respectively, in 1 M HCl electrolyte. Na_2_SO_3_ was initially added to reduce the As^5+^ to As^3+^, which induced the electrochemical response. The optimum deposition potential for As^3+^ was −0.15 V (potential window = −0.1 to 0.5 V) with a linear response between 0.133–534 pM and an LOD of 0.067 pM. The interference effect of Cu^2+^ was established along with real samples investigations. Based on LOD and applicability, it can be accounted as an impressive work, but information on measurements and sensitivity details are still missing.

Electrodeposited gold nanoparticles on a boron-doped diamond (AuNP/BDD; AuNP size = 70–90 nm) electrode was fabricated for discrimination of As^3+^ by means of square wave anodic stripping voltammetry (SWASV) response [68]. By using thiosulfate in 1.0 mol L^−1^ HCl, the As^5+^ cation was reduced to As^3+^, which gave an SWASV response in a potential window of −0.25 and 0.35 V (screen-printed carbon and Ag/AgCl were used as counter and reference electrodes, respectively) with the best result at approximately 0.05 V. The SWASV response of As^3+^ on the AuNP/BDD electrode using a multistep paper-based analytical device (mPAD) displayed a linear response between 1.33–20 nM with an LOD of 13.35 nM. This work was verified by interference study with Cu^2+^ and also engaged in determination of As^3+^ in rice samples. The obtained results agreed with that of the inductively coupled plasma-optical emission spectroscopy (ICP-OES) data. Apart from the lack of information on the B-atom concentrations, this mPAD device showed excellent performance in terms of its applicability and LODs. Shortly after, Fauzillah et al. also reported utilization of AuNPs–BDD electrodes (B/C = 10^4^ ppm; Ag/AgCl and Pt-wire act as referenced and counter electrodes, correspondingly, in 0.1 M HCl electrolyte) for quantitation of As^3+^ by anodic stripping voltammetry (ASV; potential window = −0.1 to 0.8 V; deposition potential = −500 mV and deposition time = 120 s) responses [69], wherein the AuNPs (size = 29 ± 5 nm) applied on the BDD electrode were synthesized by engaging allyl to conjugate to BDD. The AuNPs–BDD electrode displayed a linear response from 0–100 µM with an LOD of 64 nM; however, this report lacked mechanistic aspect, real-time applicability, and interference studies.

Other than the iridium-implanted BDD [66], stable iridium-modified boron-doped diamond (stable Ir-BDD; B/C = 0.1%) electrode was also employed in the detection of As^3+^ [70]. The catalytic arsenic oxidation by the electrodeposited Ir particles is described as follows.
Ir^4+^ (reduced state) → Ir^6+^ (oxidized state)(1)
Ir^6+^ (oxidized state) + As^3+^ → Ir^4+^ (reduced state) + As^5+^(2)

In Equations (1) and (2), the stable Ir-BDD displays a linear cyclic voltammetry (CV) response (potential window = −0.1–1 V; spiral Pt and Ag/AgCl acted as counter and reference electrodes, respectively, in phosphate buffer at pH 3; scan rate 50 mV s^−1^) to As^3+^ between 0–100 µM with an LOD of 4.64 µM, as shown in Figure 4. It can be stated as nice work in terms of its interference study with Cu^2+^ and investigations in real water samples.

Sugitani and coworkers described the utilization of BDD electrode (B/C = 1%) for electrochemical quantification of Cd^2+^ via the ASV response (scan range = −0.9–0.5 V; Pt-wire and Ag/AgCl acted as counter and reference electrodes, correspondingly, in 0.1 M HClO_4_) [71]. The depletion potential for Cd^2+^ was found at −0.3 V at 5 min deposition time with a linear response between 0–0.1 mM and an LOD of 3.94 nM. This work studied the interference from Cu^2+^, but more interrogations in real-time applications are still required. Zhang et al. demonstrated the consumption of BDD electrode with an 8000 ppm B atom doping concentration towards Cd^2+^ detection by means of SWASV responses (potential window = −1.0–0.0 V; saturated calomel electrode (SCE) and Pt-wire acted as reference and counter electrodes in 0.1 M HAc-NaAc buffer solution at pH 4.67; deposition potential = −1.4 V; deposition time = 500 s; scan rate = 50 mV s^−1^) [72]. The electrode showed a linear response from 0.18–2.17 µM with an LOD of 8.8 nM. Note that the interference effect was negligible with other metallic species and the results were reproducible with real samples and were comparable with other electrodes, thereby can be accounted an impressive work.

Thereafter, Innuphat et al. demonstrated the use of 4-aminomethyl benzoic acid modified BDD electrode for determination of Cd^2+^ via SWASV responses (potential window = −0.2 to −1.0 V; Ag/AgCl and Pt were used as reference and counter electrodes in acetate buffer at pH 6; deposition potential = −0.72 V; deposition time = 6 min; scan rate = 100 mV s^−1^) [73]. The diazonium salt was formed and grafted over the BDD electrode by the amino group of 4-aminomethyl benzoic acid in the electrochemical cell, which played the mechanistic role of detecting Cd^2+^. The electrode displayed a linear response to Cd^2+^ from 18 to 445 pM with an LOD of 1.8 pM and was effective in the presence of interfering ions. Applicability of the detection process in real and NIST samples agreed with the ICP-OES analysis, which was noted as an added advantage, but the doping level of B atom must be clarified before commercialization.

The fabricated BDD electrodes can be applied in multiple analyte detection, such as neurotransmitter and heavy metal ions, as demonstrated by Nantaphol and coworkers [74], wherein the BDD (B atoms at 10^20^–10^21^ cm^−3^ concentration) paste electrode coupled with a microfluidic paper-based analytical device (μPADs) was employed in detection of serotonin and norepinephrine as well as Cd^2+^ and Pb^2+^. By means of SWASV responses (potential window = −1.1 to 0.5 V; Ag/AgCl on a transparency sheet used as a conducting pad in in 0.1 M acetate buffer (pH 4.5)), both Cd^2+^ and Pb^2+^ detections were conducted. On the other hand, serotonin and norepinephrine detections were conducted by CV studies. Linear responses for Cd^2+^ and Pb^2+^ were established as 8.9 pM–1.78 nM and 4.83 pM–0.965 nM with LODs of 0.22 nM and 4.83 pM, respectively. In terms of the interference and drinking water studies, this design was exceptional. Pei et al. proposed the replacement of mercury film electrode by BDD with B/C ratio of 2.4% towards simultaneous detection of Cd^2+^ and Pb^2+^ [75], wherein the sensing effect of hydrogen-terminated electrode (H-BDD) and oxygen-terminated electrode (O-BDD) was demonstrated by SWASV studies (scan range = +0.8 to −0.8 V; SCE and Pt were used as reference and counter electrodes; enrichment time of 200 s; scan speed 50 mV s^−1^; NaAc-HAc buffer at pH 4.68). The H-BDD showed greater sensitivity to Cd^2+^ and Pb^2+^ with linear correlations of 0.05–4 µM and 0.05–8 µM and LODs of 30.16 and 17.47 nM, respectively. Moreover, the spike recovery investigations also authenticated the suitability of the electrode.

To this track, diverse structured and graphitized BDD electrodes (as grown-BDD, BDD nanotips, small and larger nanorods) were engaged in determination of Cd^2+^ and Pb^2+^ by Štenclová and coworkers [76]. This report discussed the importance of structural and sp^2^ graphitic shells in the detection process by DPASV responses (scan window = −1.0 to 0.1 V; Ag/AgCl and Pt-wire as reference and counter electrodes in 0.1 M HCl; accumulation time = 60 s). Based on their results, it was concluded that small nanorods and BDD-nanotips possess the higher electrochemical performance to Cd^2+^ (DPASV peak at −0.77 to −0.78 V) and Pb^2+^ (DPASV peak at −0.49 to −0.51 V). In particular, the small-nanorods-structured BDD electrode has a suitable morphology to deliver the separate response to Cd^2+^ and Pb^2+^. However, information on linear regression and LODs were not clear enough to authenticate the electrodes performance. In a similar fashion, diamond/carbon nanowalls film (D/CNWs—3%, D/CNWs—5%, D/CNWs—7% and D/CNWs—9%) electrodes were proposed for detection of Cd^2+^ and Pb^2+^ through the DPASV responses [77]. The involvement of graphite sp^2^-C in the electrochemical sensing process was illustrated. Investigations on the D/CNWs—3% electrode (sp^2^/sp^3^ ratio = 1.38; scan range = −0.99 to 1.73 V; Ag/AgCl and Pt-wire as the reference and counter electrodes in 0.1 M H_2_SO_4_ scan rate = 100 mV s^−1^) show linear regressions of 0.089–8.9 µM (Cd^2+^) and 0.048–4.83 µM (Pb^2+^) with LODs of 89 nM (Cd^2+^) and 48 nM (Pb^2+^), respectively, as depicted in Figure 5. Finally, this report establishes the involvement of sp^2^-C in sensing of Cd^2+^ and Pb^2+^, but real-time application and interference studies are still required.

To quantify the Cr^6+^, Fierro et al. engaged the BDD electrode (B/C = 0.1%) by linear scan voltammetry (LSV) responses (scan range = 0.6 and −0.3 V; Ag/AgCl and Pt-wire as reference and counter electrodes in 0.1 M HNO_3_; scan rate = 50 mV s^−1^) [78]. As seen in Figure 6, a linear response of BDD to Cr^6+^ is estimated as 0.2–96 nM with an LOD of 0.57 pM. This work was also demonstrated with interference studies, but detection mechanism and real-time application needs to be updated.

Recently, cathodically pretreated AuNPs–BDD electrode (AuNPs size was unclear) was utilized in electrochemical detection of Cr^6+^ by Xu and coworkers [79]. Square wave cathodic stripping voltammetry (SWCSV) responses of the AuNPs–BDD electrode to Cr^6+^ (scan range = 0.6 to 0 V; Ag/AgCl and Pt-wire as the reference and counter electrodes in 0.1 M sodium acetate buffer at pH 6; AuNPs deposition time 300 s; scan rate = 50 mV s^−1^) showed a linear response from 0.193 to 19.3 µM and an LOD of 22.91 nM. The AuNPs-enhanced adsorption of Cr^6+^ over the electrode undergoes reduction to produce Cr^3+^, as shown in Figure 7, which results in the electrochemical response. This was also demonstrated with interference effects, real sample analysis, and was comparable to other existing electrodes.

Nie et al. described the detection of Cu^2+^ and proposed marine corrosion monitoring by using the BDD disk electrode (B atom at 2000 ppm doping level) [80]. In the presence of Cu^2+^, the BDD electrode displayed a linear differential pulse voltammetry (DPV) response (scan range = −0.1 to 0.8 V; Ag/AgCl and graphite rod as reference and counter electrodes in 0.6 M NaCl; scan rate = 10 mV s^−1^) from 10 µM–100 mM with an LOD of 10 µM. During the detection process, the Cu^2+^ reduced to Cu^+^ and resulted in electrochemical response. The apparent rate constant for the Cu^2+^/Cu^+^ redox process (quasireversible process) in chloride electrolyte was estimated as 0.94 × 10^−6^ cm s^−1^. Therefore, it can be used for scrutinization of marine corrosion samples. Moreover, this electrode shows negligible interference with other ions; therefore, it is reliable for Cu^2+^ detection.

The BDD electrode was also engaged in detecting toxic Hg^2+^, as detailed next. Manivannan et al. utilized the BDD electrode to discriminate Hg^2+^ via DPV responses (scan range = −0.1 to 0.5 V; SCE and Pt- reference and counter electrodes in 0.1 M KNO_3_ (pH 1); deposition time = 20 min), which showed a linear response of 10 nM–0 µM with an LOD of 68 nM [81]. However, this work lacks interference and real analysis; therefore, it can be noted as a preliminary study. Thereafter, McLaughlin and coworkers demonstrated the use of AuNP-decorated BDD (B atoms at 10^20^ cm^−3^ doping concentration) towards detection of Hg^2+^ via SWASV responses and stripping voltammetry-based electron impedance spectroscopic (EIS) studies [82,83]. The SWASV response of AuNPs–BDD to Hg^2+^ (scan range = 0.35 to 1.1 V; Ag/AgCl and Pt-wire as the reference and counter electrodes in 3M KCl; scan rate = 2.5 mV s^−1^) displayed a linear response from 0.5–100 µM with an LOD of 5 µM [82]. On the other hand, the SWASV-based EIS studies under the similar condition displayed a wide linear regression from 1 pM to 1 mM [83]. In both reports, authors explained the role of the AuNPs in different sizes (22 nm and 30 nm, respectively) and the sp^2^/sp^3^ carbon ratios in the enhanced response to Hg^2+^. It was concluded that the fabricated electrode was able to detect Hg^2+^ even at picomolar level.

The BDD electrode (10^20^ cm^−3^ doping of B atom)-mediated quantification of Ni^2+^ in neutral or acidic media was justified by Neodo and coworkers [84] wherein the BDD shows a linear DPV response (scan = −1.1 to 1.3 V; range = Ag/AgCl and Pt-wire as reference and counter electrodes in phosphate or 0.6 M NaCl; deposition potential of Ni^2+^ = 30 s; scan rate = 10 mV s^−1^) from 10–500 µM with an LOD of 26.1 µM in the presence of Ni^2+^. This work provided valuable information in the interference and cleaning process, but the real-time applications were still missing. Musyarofah’s and Yuliani’s research groups proposed the determination of Ni^2+^/Ni(OH)_2_ NPs by using BDD electrodes in separated reports, wherein the BDD exhibited linear ASV responses (Ag/AgCl and Pt-wire as the reference and counter electrodes in 0.1 M PBS/0.1 M HClO_4_ at pH 3; deposition time of 300 s/90 s; scan rate = 100 mV s^−1^) between 5–25 mM (in both reports) with LODs of 5.73 and 0.42 µM, respectively [85,86]. Both reports did not disclose information on the B-doping level, interference, and real analysis.

Detection of Pb^2+^ was also demonstrated with the BDD electrodes, as described in many reports [87,88,89,90]. Those BDD electrodes (Ag/AgCl or SCE and Pt-wire/coil act as the reference and counter electrodes) have revealed the B/C ratio-induced linear SWASV responses and LODs at nanomolar/picomolar levels. In particular, Pei et al. reported the utility of self-supported boron-doped diamond (SBDD) electrode towards Pb^2+^ quantification [90]. The double side effect of B/C ratio on the electrochemical performance of SBDD was also discussed in this report. As shown in Figure 8, amount of both B–sp^3^–C phase and D(111) grains are found to be greater at a B/C ratio of 1/500 (Stage II); therefore, the number of active sites present in the SBDD electrode are increased to adsorb Pb^2+^ as a result of higher sensor responses. On the other hand, the sensory responses were not significant due to lack of the electron injection at B/C ratios of 1/2500 (Stage I) and 1/250 (Stage III). By means of square wave anodic stripping voltammetry (SWASV between −0.1 V to −0.5 V; SCE and Pt were used as reference and counter electrodes in 0.1 M Na_2_SO_4_ at pH 4.68; between 225–300 s, Pb^2+^ accumulation on SBDD was found to be higher), a linear response of SBDD to Pb^2+^ was established from 15–362 nM with an LOD of 3.38 nM. Note that the SBDD electrode displayed sensitivity of 0.42 μA L μg^−1^ cm^−2^, thereby becoming an unique material in electrochemical sensing research. This work demonstrates the importance of B–sp^3^–C phase and electrode grain morphology in electroanalytical studies, but real-time applications are still missing.

One-dimensional, nitrogen-doped diamond nanorods (N-DNRs) and unmodified BDD (B = 8000 ppm) were engaged in the discrimination of Pb^2+^/Cd^2+^ and Pb^2+^/Cu^2+^, respectively [91,92]. The N-DNRs-based electrode revealed linear SWASV responses (Ag/AgCl and Pt-wire as reference and counter electrodes in acidic media) from 50 nM to 1 µM and 10 nM to 1.1 µM with LODs of 50 and 10 nM for Pb^2+^ and Cd^2+^, correspondingly. The enhanced electrochemical response was attributed to the improved electrical conductivity in the grain boundary regions by the sp^2^ nanographitic phases. On the one hand, the unmodified BDD electrode was engaged in simultaneous detection of Pb^2+^ and Cu^2+^ via the SWASV responses (scan range = + 0.20 to −1.20 V; SCE and Pt as the reference and counter electrodes; 210 s (deposition time)), which showed linear range of 30–180 nM (for both Pb^2+^ and Cu^2+^) with LODs of 27 and 4 nM, correspondingly [65]. Note that this electrode displays good responses over many interferences and is also applied in biodiesel studies.

Voltametric determination of Sb^3+^ was demonstrated by the cathodically pretreated BDD electrode (B = 1000 ppm or 10^20^ cm^−3^) via eliminating interfering effect of As^3+^ by using NaH_2_PO_4_ as the supporting electrolyte and EDTA as the selective complexing agent for Sb^3+^ [93]. As shown in Figure 9, the BDD electrode shows a linear DPASV response (Ag/AgCl and Pt as the reference and counter electrodes in 6 M HClO_4_; deposition potential = −1 V; deposition time = 240 s) from 2.44 to 7.31 µM with an LOD of 108 nM. Based on the interference and real analysis, this work is considered remarkable.

Electrochemical discrimination of Zn^2+^ by the BDD electrode was authenticated by Culková and coworkers [94]. The electrode displayed a linear DPASV response (scan range = −1.1 to −1.7 V; Ag/AgCl and Pt-wire as the reference and counter electrodes in 0.1 M KCl; deposition time = 120 s) from 0.5 nM–5 µM with an LOD of 0.47 nM. This work delivered important information in the linear regression, interference, and rubber industry samples analysis.

Simultaneous quantitation of three (and more) heavy metal ions were demonstrated by many nanodiamond-based electrodes, such as nanocrystalline diamond (NCD), polyphenol–polyvinyl chloride-modified boron-doped diamond (PVC-BDD), boron-doped nanocrystalline diamond (BD-NCD), boron-doped diamond (BDD), bismuth-modified BDD, and diamond/graphite nanoplatelets electrodes, via LSV, ASV, DPASV, and SWASV responses [95,96,97,98,99,100,101,102,103]. Note that the electrochemical performances of those electrodes are attributed to the graphitic phase generation and sp^2^/sp^3^ ratio-tuned conductivity enhancement. In all of the studies, either Ag/AgCl or SCE and Pt or graphite rods were utilized as the reference and counter electrodes at suitable pH values. The BDD electrodes displayed exceptional results in simultaneous detection of multiple metal ions. In particular, the bismuth-modified BDD electrode [97] reveals simultaneous SWASV responses to Zn^2+^, Cd^2+^, and Pb^2+^, as shown in Figure 10. It indicates that the B atomic concentration and surface morphology play vital roles in the electrochemical investigations.

Thereafter, Zhai et al. engaged the diamond/graphite nanoplatelets electrode for simultaneous quantification of Zn^2+^, Cd^2+^, Pb^2+^, and Cu^2+^ via improved conductivity by introducing graphitic shells into the diamond phase [103]. The diamond/graphite nanoplatelets electrode shows exceptional simultaneous linear responses to Zn^2+^, Cd^2+^, Pb^2+^, and Cu^2+^, as seen in Figure 11. Many of the electrodes are able to simultaneously detect multiple ions but with certain complications. Thus, careful optimization becomes essential. Table 1 summarizes the doping concentrations of specific atoms, methods of detection, linear regressions, and LODs of nanodiamond-based electrodes towards metal ions [62,63,64,65,66,67,68,69,70,71,72,73,74,75,76,77,78,79,80,81,82,83,84,85,86,87,88,89,90,91,92,93,94,95,96,97,98,99,100,101,102,103].

## 3. Diamond-Based Electrodes in Anions Discrimination

Due to the environmental issue, detection and quantification of specific anions by diamond-based electrodes are also described in this section. Xu and coworkers demonstrated the irreversible oxidation of azide (N^3−^) anion at the BDD-thin film electrode (B at 1 × 10^19^ cm^−3^ concentration) and made comparisons with the glassy carbon electrode [104], wherein the following oxidation reaction leads to the electrochemical signal:2N^3−^ → 3N_2_ + 2e(3)

The BDD electrode displayed responses to N^3−^ via LSV, DPV, and flow injection analysis (SCE and Pt-wire as reference and counter electrodes in 0.1 M phosphate buffer at pH 7.2; scan rate = 50 or 100 mV s^−1^) with a linear response from 3.3 mM–0.30 µM and an LOD of 8 nM. It is a pioneer work towards N^3−^ quantitation. To quantify the peroxide (O_2_^2−^) anion, Pt-implanted and Prussian blue-modified BDD electrodes (Pt-BDD and BDD/PB) were engaged in the electrochemical assay of hydrogen peroxide (H_2_O_2_) via CV and flow injection analysis [105,106]. The BDD (B/C = 1:100) electrode was implanted with 50 keV Pt^2+^ with a dose of 5 × 10^14^ cm^−2^. By using Ag/AgCl and Pt-wire as the reference and counter electrodes in 0.1 M phosphate buffer at pH 7, the H_2_O_2_ oxidation at Pt-BDD was compared with Pt electrode [105], wherein the BDD electrode showed a linear response to H_2_O_2_ from 0.1 to 10 µM with an LOD of 30 nM and was applicable in flow injection analysis. Next, the Prussian blue was electrodeposited over H-terminated BDD electrode to form the BDD/PB electrode, which showed sensitivity of 0.14 A M^−1^ cm^−2^ to H_2_O_2_ (Ag/AgCl and Pt-wire or graphite rod acted as the reference and counter electrodes in 0.05 M phosphate buffer at pH 6). The results were superior compared to the conventional graphite-based electrode [106]. With respect to sensitivity and applicability, both reports are inspiring towards biological studies. In this course, the BDD electrode was also reported for in situ production of coreactant H_2_O_2_ in carbonate (CO_3_^2−^) aqueous solution by Einaga et al. [107], wherein the H_2_O_2_ production was demonstrated through chemiluminescence signal.

The diamond paste electrode was utilized as a reliable technique towards the electrochemical assay of iodide (I^−^) anion [108]. The diamond paste electrode displayed a linear DPV response (potential range = +0.5 to −0.7 V; Ag/AgCl (in 0.1 M KCl) and Pt acted as reference and counter electrodes; scan rate = 25 mV s^−1^) between pM to nM with an LOD at subnanomolar level. Although this work showed applicability in vitamins and table salt, many details on interference and other data are missing. Fierro and coworkers demonstrated the use of BDD (0.1% wt B atom) electrode for the quantitation of I_2_ and I^−^ in 1 M NaClO_4_ (pH 8) solution [109]. The I^−^ from KI solution in BDD electrode underwent oxidation to form I_2_ and then was further oxidized to iodate (IO_3_^−^) ions, but this technique is applicable to I^−^ and I_2_ detection only. From the CV measurements (potential range = 0 to 2.5 V; Ag/AgCl and Pt-wire were used as the reference and counter electrodes in 1 M NaClO_4_ at pH 8; scan rate = 100 mV s^−1^), linear responses for I^−^/I_2_ were found between 0–1.2 mM/0–0.6 mM with LODs of 10 µM/20 µM. This is a unique work, but further updates on the interference and real-time applications are still required. The BDD electrode was also employed for indirect estimation of fluoride (F^−^) anion via the formation of electrode-mediated electroinactive fluoride complexes {[FeF_6_]^3−^ and [CeF_6_]^2−^} with Fe^3+^ and Ce^6+^ [110]. Drinking water containing F^−^ was investigated by the BDD electrode in the mixture of 1 mM FeCl_3_ in 0.2 M NaCl or 1 mM Ce(SO_4_)_2_ in o.1 M H_2_SO_4_. The reactions led to the electrode-mediated complex formation, as represented below.
Fe^3+^ + 6F^−^ → [FeF6]^3−^(4)
Ce^4+^ + 6F^−^ → [CeF6]^2−^(5)

From LSV and square wave voltammetry (SWV) responses (Ag/AgCl and Pt as the reference and counter electrodes), the LODs for F^−^ estimation via [FeF_6_]^3−^ and [CeF_6_]^2−^ complexes were calculated as 5 µM and 0.6 µM, correspondingly. This is a good research report, which utilizes the Fe^3+^/Fe^2+^ and Ce^4+^/Ce^3+^ redox process for detection of F^−^.

Thereafter, Lucio et al. disclosed the use of sp^2^ carbon-inserted BDD electrodes in quantification of hypochlorite (OCl^−^) [111,112] and in estimation of chloride ions. As seen in Figure 12, the sp^2^ carbon-inserted BDD electrode (B = 10^20^ atoms cm^−3^) shows a linear SWV response (SCE and coiled Pt-wire as the reference and counter electrodes; scan rate = 40 mV s^−1^) from 0 to 1.50 M for chloride estimation [111]. The sp^2^-BDD electrode was effective for OCl^−^ detection in pH range from 4 to 10. On the one hand, the sp^2^-bonded carbon microspot-BDD (b = 10^20^ atoms cm^−3^) electrode [112] is utilized in the voltametric (by LSV and SWV signals) detection (SCE and Pt as the reference and counter electrodes) of both OCl^−^ and pH, as represented in Figure 13. Moreover, these two reports clearly demonstrate the use of sp^2^ graphitic shells-inserted BDD electrode towards OCl^−^ and Cl_2_ quantitation at pH 4–10.

The OCl^−^ undergoes voltametric oxidation reaction in water, as described in Equation (6), which results in electrochemical signals.
6OCl^−^ + 3H_2_O → 2ClO_3_^−^ + 4Cl^−^ + 6H^+^ + 3/2O_2_ + 6e^−^(6)

Note that the sp^2^-bonded carbon microspot-BDD electrode showed a linear LSV response from 58.31 µM to 1.9 mM with an LOD of 58.31 µM.

Juliao et al. described the BDD electrode (B doping at 10^21^ cm^−3^ level) towards voltametric determination of interactions between RNO_2_^•−^ and electron acceptors existing in nitrofurazone (NFZ) in aqueous phase [113]. This highly B-doped diamond electrode displayed linear DPV responses (scan range = −0.2 to −0.6 V; Ag/AgCl/KCl and Pt-wire acted as the reference and counter electrodes in BR buffer at pH 4 and 8; scan rate = 100 mV s^−1^) from 0.99 to 17 µM (in the absence of O_2_ at pH 4) and 0.99 to 11 µM (in the presence of O_2_ at pH 8) with LODs of 0.41 and 0.34 µM, respectively. This work provided reliable detection tactic towards the NFZ derivatives. Thereafter, electrochemical sensing of nitrite (NO_2_^−^) by using modified BDD electrode (B = 7000–8000 ppm) was proposed by Sahraoui and coworkers [114], wherein p-phenylenediamine (PPD) and silicotungstate polyoxoanion (SiW_11_) were uniformly formed over the BDD microcells electrode surface via layer-by-layer assembly to provide the final structure of BDD/PPD/SiW_11_, which displayed a stronger anodic CV response to NO_2_^−^ at −0.6 V in 0.1 M H_2_SO_4_ (scan rate = 100 mV s^−1^).

In this report, a pseudoreference made of BDD was engaged to replace the conventional SCE reference electrode, and SiW_11_ acted as a mediator for selective detection of NO_2_^−^. From the SWV investigations, the BDD/PPD/SiW_11_ device displayed a linear response from 4 µM–4 mM with an LOD of 20 µM. This work can be attested to as a nice study in terms of its device structure, interference, and real river-water samples interrogations.

More recently, Triana et al. proposed the use of BDD electrode (B/C = 1%) towards the electrochemical detection of nitrous acid (HONO) and NO_2_^−^ [115], wherein BDD, GC, Pt, and stainless steel were employed as the working electrodes with 1% BDD and Ag/AgCl (saturated KCl) acting as the counter and referenced electrodes, respectively. The following oxidation reactions of HONO and NO_2_^−^ take place during the electrochemical detection process.
HONO + H_2_O → NO^•^_3_ + 3H_+_ + 3e^−^(7)
NO_2_^−^ + H_2_O → NO_3_ + 2H^+^ + 2e^−^(8)

The electrode showed linear CV responses to HONO and NO_2_^−^ (scan range = 0 to 2 V in 0.1 M KClO_4_; scan rate =100 mV s^−1^) from 1 to 5 mM (for both) with the LODs of 0.24 and 1.27 nM, correspondingly. Figure 14 shows the CV response of BDD electrode towards various concentrations of NO_2_^−^ anions. The analytical performance of the BDD electrode is comparable with earlier reports.

In situ CuNPs-deposited BDD electrode was explored in determination of NO_3_^−^ by Welch and coworkers [116]. Unknown boron doping concentration and 100 µM of Cu^2+^ in phosphate buffer was used for electrodeposition over the BDD electrode. The electrode displayed a linear LSV response (scan range = −0.2 to −1.2 V; SCE and Pt as the reference and counter electrodes in 0.1 M Na_2_SO_4_ at pH 3; scan rate = 10 mV s^−1^) from 0 to 100 µM with an LOD of 1.5 µM. This research gave a good report in the real water analysis but need more work on optimization of Cu^2+^. Thereafter, Kuang et al. reported the effect of surface termination and boron doping level on electrochemical reduction of nitrate (NO_3_^−^) [117]. This report discussed diverse boron doping levels on BDD electrodes (B/C = 0.1, 1, 2, and 3%) and revealed LSV responses (scan range = −1.0 to 2.0 V; Ag/AgCl (saturated KCl) and Pt acted as the reference and counter electrodes in 0.1 M HClO_4_) to different concentrations of NO_3_^−^. It is a novel method towards the NO_3_^−^ detection. Electrogeneration of peroxydisulfate (S_2_O_8_^2−^) from the sulfate (SO_4_^2−^) anions using the BDD electrodes was also reported by researchers [118,119]. Therefore, the BDD electrodes can be employed to estimate the S_2_O_8_^2−^ anion. Subsequently, Kondo and coworkers demonstrated the surface-modified BDD electrode (B/C = 10,000 ppm) towards detection of the oxalate dianion (C_2_O_4_^2−^) [120]. The BDD surface becomes positively charged by covalently attaching the allyltriethylammonium bromide (ATAB) over the H-terminated BDD surface to deliver the final structure of ATAB-BDD. The electrode displayed an amperometric response to C_2_O_4_^2−^ (Ag/AgCl and Pt as the reference and counter electrodes) from 0.8–100 µM with an LOD of 32 nM. Moreover, the BDD electrode was employed in the amperometric discrimination of the sulfide (S^2−^) anion via the electrocatalytic reaction with ferricyanide in aqueous solution [121], wherein the electrode responded to the S^2−^ anion linearly between 8–43 µM with an LOD of 3 µM. This work demonstrates a notable tactic in determining the effect of ferricyanide in S^2−^ detection. Similarly, the BDD electrodes were also engaged in simultaneous detection and electrogeneration of two (and more) anionic species [122,123,124,125], thereby becoming a unique candidate for electrochemical analysis. Table 2 summarizes the doping concentration of specific atoms, methods of detection, linear regressions, and LODs of nanodiamond-based electrodes towards anions [104,105,106,107,108,109,110,111,112,113,114,115,116,117,118,119,120,121,122,123,124,125].

## 4. Optimization Requirements

Though diamond-based electrodes are promising candidates toward quantitation of metal ions and anions, optimization through the following steps is indispensable to achieve reliable results.

1)Generally speaking, synthesis of diamond-based nanomaterials is mostly carried out by CVD techniques [126], which require careful optimization in operating temperatures and chamber pressures to deliver the appropriate nanostructures and morphology.2)For the surface-tuned electrochemical detection process, it is essential to examine tiny morphology changes through microscopic techniques [127,128,129].3)In the fabrication of specific atom- or nanoparticle-doped electrodes (for example, BDD, AuNPs–BDD, Ir-BDD, and Pt-implanted BDD), cautious optimization with doping concentrations is essential to attain reproducible results [130].4)When the diamond-based electrodes, such as BDD electrodes, are used for detection, it is important to select the suitable reference (example: Ag/AgCl or SCE) and counter electrodes (example: Pt, graphite, GCE, etc.). Therefore, great attention is required in selection of the reference and counter electrodes [131].5)Film thickness and working/active surface area of electrodes must be carefully optimized and determined to attain reliable results towards specific analytes [132,133].6)To achieve repeatable results to specific analytes, the supporting electrolytes play a crucial role in enhancing the redox process that leads to the electrochemical signals [134]. Thus, optimization is required in determining the suitable supporting electrolyte to enhance the redox reaction.7)Suitable operating pH values and temperature must be optimized and fixed to enhance the analyte-specific redox process [135,136].8)Finally, optimization in maintaining highly reliable and reproducible electrochemical data signals (such as CV, ASV, DPV, LSV, DPASV, and SWASV), scan range, scan rate, etc., under the aforementioned conditions is mandatory.

## 5. Advantages

Using the diamond-based electrodes towards metal ions and anions detections has the following advantages, as stated below.

1)Due to their unique structural and electrochemical resistivity features, diamond-based electrodes can be operated effectively towards specific metal ions or anions detections in suitable aqueous media and under operable conditions.2)By tuning the B-atom doping concentration in the BDD electrode, it can be used effectively in detection and quantification of diverse metal analytes with specificity.3)Modifications of the BDD electrodes by foreign materials or metallic nanoparticles can lead to different structural and morphological features, which allow the accumulation of specific metal ions and anions and lead to electrochemical responses via redox reaction.4)By inclusion of the graphitic or nanographitic shells in the diamond-based electrodes, the conductivity can be enhanced by tuning the B/C and sp^2^/sp^3^ ratios; therefore, highly responsive signals can be obtained.5)In general, the diamond-based electrode is able to carry out detection down to picomolar level with wide linearity, which is comparable to other existing sensing tactics, such as organic probes, nanoparticles, quantum dots, nanoclusters, carbon dots, etc. [137,138,139,140,141,142,143,144].6)In terms of the lowest LODs for toxic heavy metal ions and anions, the diamond-based electrodes open up a new path to quantify those species with specificity.7)Recent reports on diamond-based electrodes demonstrate the exclusive features for simultaneous detection of multiple metal ions and anions, which are highly beneficial.

## 6. Limitations

Though diamond-based electrodes have many benefits towards metal ions and anions detections, they also possess the following limitations.

1)Analyte quantitation by the diamond-based electrodes is limited to the availability of sophisticated electrochemical instruments and laboratory environment.2)Synthesis of diamond nanostructures requires CVD tactics operated at high temperatures. Moreover, the electrode fabrication with certain film thickness is limited by many complicated procedures.3)Morphological investigations on the diamond-based electrodes require microscopic investigations, thereby limiting by the use of costly equipment, such as SEM, TEM, AFM, etc.4)Reproducibility of electrochemical signals to specific analytes is limited by the prescribed reference and counter electrodes, supporting electrolyte, pH value, and temperature conditions. Changes in the measurement conditions may result in loss of data.5)The electrochemical results obtained from the diamond-based electrodes could possibly be affected due to changes in electrode surface and loss of signal by some highly concentrated unknown interferences existing in diverse real samples.6)Use of hazardous acidic electrolytes, such as HCl, H_2_SO_4_, buffer solutions, and toxic Hg/Hg_2_Cl_2_ electrode, in the detection process is harmful to the environment, thereby limiting the real-time electrochemical interrogations.7)Though the diamond-based electrodes are able to detect metal ions and anions down to picomolar level, their biological applicability still needs to be validated in many cases.

## 7. Conclusions and Perspectives

This review outlines the consumption of diamond-based electrodes in detection and quantification of metal ions and anions via reliable redox processes. Involvement of the surface and morphological changes and graphitic shells in the redox process in the enhanced electrode conductivity to provide stronger electrochemical signals are clarified for the readers. The effect of boron-doping concentration and sp^2^/sp^3^ ratio in improving the nanographitic shells on BDD electrodes to expand the signals towards specific metal ions and anions detection has been described briefly. Modifications of diamond or BDD electrodes by metallic implantation and nanoparticles coating to deliver higher electrochemical signals have also been detailed in this review. Information on the reference electrodes, counter electrodes, supporting electrolytes, scan range, and scan rates are provided, along with the tabulation of doping atom concentration, method of detection linear range, and LODs. However, the diamond-based electrodes can only be commercialized for discrimination of metallic and anionic species if the following perspective points are addressed.

1)The diamond/BDD electrodes are typically fabricated by high-temperature CVD techniques. Thus, alternative wet-chemical routes must be established for easy synthesis and cost-effective commercialization.2)Though reports on the diamond/BDD-based electrodes for detecting the metal ions and anions are impressive, currently there is no “state-of the art” procedure for commercialization.3)Some reports did not provide information on the B-doping concentration, particle size modified over the electrode surface, and the effectiveness of the doping and modification on electrochemical interrogations. This should be rectified in upcoming research reports.4)Similar to the case of implantation and metallic nanoparticles modifications, the diamond/BDD electrodes can be modified by certain emerging materials, such as MOFs and COFs, to further tune the detection signals towards specific metal ions and anions. However, this requires careful optimization procedure.5)A few reports on the metal ions detection did not provide proper explanations regarding the redox process, suitable electrolytes, sensitivity, role of pH, temperature, and information on limitations. The missing information still requires further clarification.6)The use of toxic SCE electrodes, highly concentrated acidic electrolytes, and buffer solution must be reduced to become truly environmentally friendly.7)Demonstrations of the diamond/BDD-based electrodes towards quantification of Co^2+^, Mn^2+^, Pd^2+^, Ru^3+^, Al^3+^, Ga^3+^, In^3+^, etc., are not yet available, which should be addressed by upcoming researchers.8)Demonstrations of the diamond/BDD-based electrodes towards electrochemical detection of lanthanides and actinides must be launched to increase the social impact.9)Some reports on the diamond/BDD-based electrodes towards anionic species detection did not have any clear information on the electron transport or generation information, which require more evidence to understand the underlying mechanisms.10)Reports on the diamond/BDD-based electrodes for anions detection are still insufficient, thereby requiring more research work toward this direction.11)Detection of anions, such as Br^−^, CN^−^, ClO_4_^−^, H_2_PO_4_^−^, S_2_O_3_^2−^, CO_3_^2−^, PO_4_^3−^, P_2_O_7_^4−^, etc., by the diamond/BDD-based electrodes has not yet been reported, which should be the focus in future.

Apart from above viewpoints, the diamond-based electrodes for metal ions and anions detection showed unique electrochemical performance with wide linear ranges and picomolar level in LODs; therefore, this research tactic is highly regarded.

## Figures and Tables

**Figure 1 nanomaterials-12-00064-f001:**
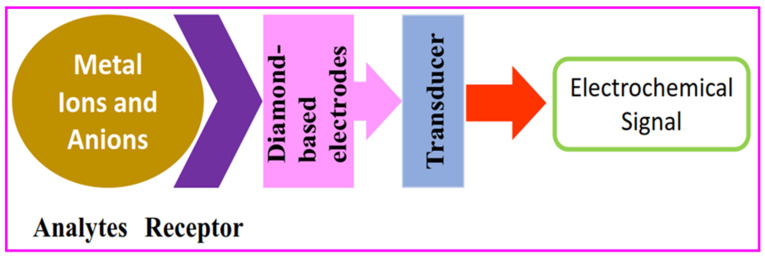
General representation of electrochemical detection of metal ions and anions by diamond-based electrodes.

**Figure 2 nanomaterials-12-00064-f002:**
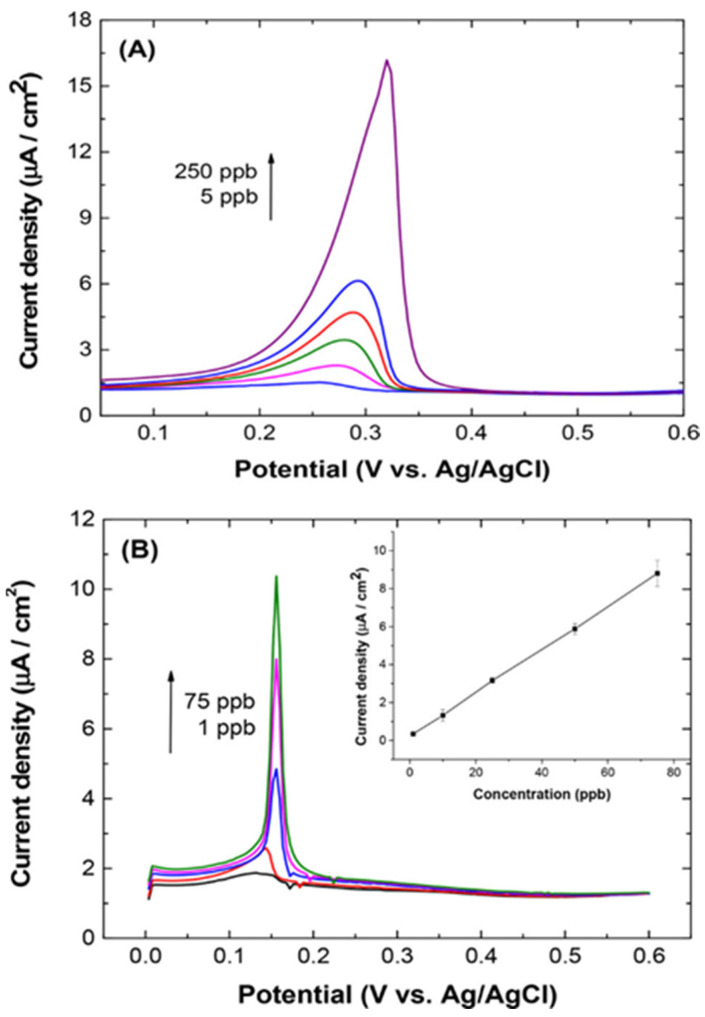
Differential pulse anodic stripping voltametric (DPASV) I−E curves for standard solutions (1−250 μg L^−1^) of Ag(I) in 0.1 mol L^−1^ acetate buffer (pH 4.6) at (**A**) planar film and (**B**) BDD disk electrodes. The deposition potential was −0.3 V and the deposition time was 120 s. The curves were recorded using a pulse amplitude of 0.05 V; a potential step of 0.004 V; a pulse width of 50 ms; and a pulse period of 100 ms. Concentrations are shown as ppb (μg L^−1^) (reproduced with the permission of [62]).

**Figure 3 nanomaterials-12-00064-f003:**
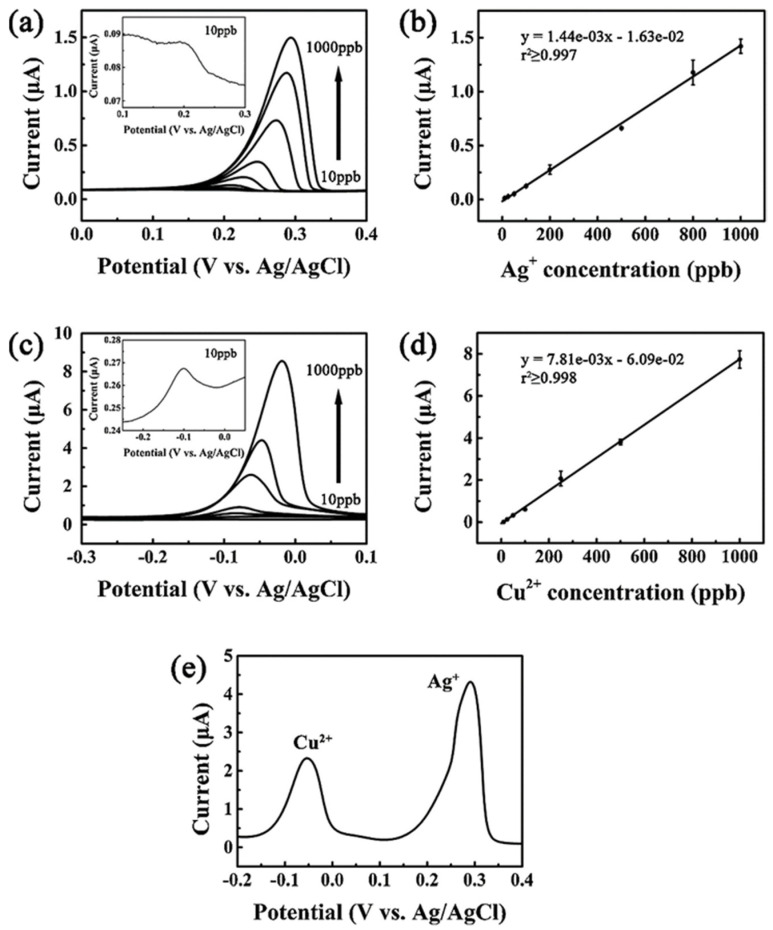
Anodic stripping voltammograms of (**a**) silver and (**c**) copper in heavy metal ion detection. The ion concentrations in silver standard solutions are 10, 25, 50, 100, 200, 500, 800, and 1000 ppb. Deposition potential: −0.1 V; deposition time: 3 min. The ion concentrations in copper standard solutions are 10, 25, 50, 100, 250, 500, and 1000 ppb. Deposition potential: −0.4 V; deposition time: 3 min. Calibration plots for Ag^+^ and Cu^2+^ are shown in (**b**) and (**d**). The error bars correspond to the standard deviation are obtained from five measurements (n = 5). (**e**) The simultaneous determination of silver and copper ions in aqueous solutions. The scan rate is 20 mV s^−1^ (reproduced with the permission of [64]).

**Figure 4 nanomaterials-12-00064-f004:**
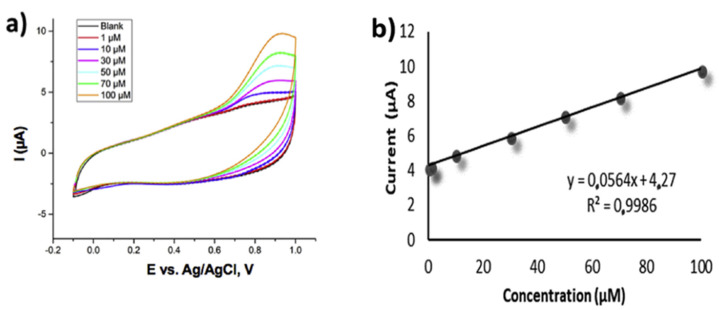
(**a**) CV responses of different concentrations of arsenic (III) in phosphate buffer solution pH 3; scan rate 50 mV s^−1^ at Ir-BDD prepared using complete step deposition. (**b**) Graph depicts the dependence of current responses on arsenic (III) concentrations (reproduced with the permission of [70]).

**Figure 5 nanomaterials-12-00064-f005:**
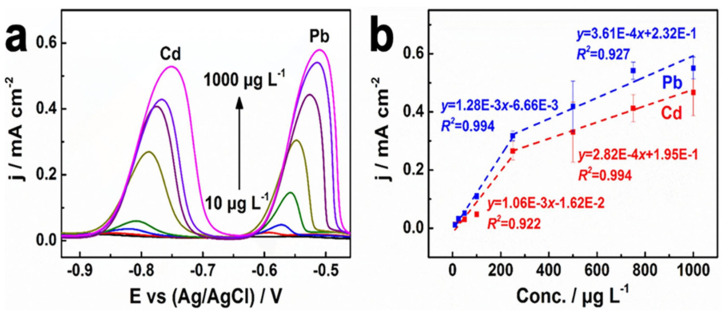
(**a**) Differential pulse anodic stripping voltammograms and (**b**) corresponding calibration plots for simultaneous determination of Cd^2+^ and Pb^2+^ with concentrations of 10, 25, 50, 100, 250, 500, 750, and 1000 µg L^−1^ on D/CNWs—3% electrode. Error bar: n = 3 (reproduced with the permission of [77]).

**Figure 6 nanomaterials-12-00064-f006:**
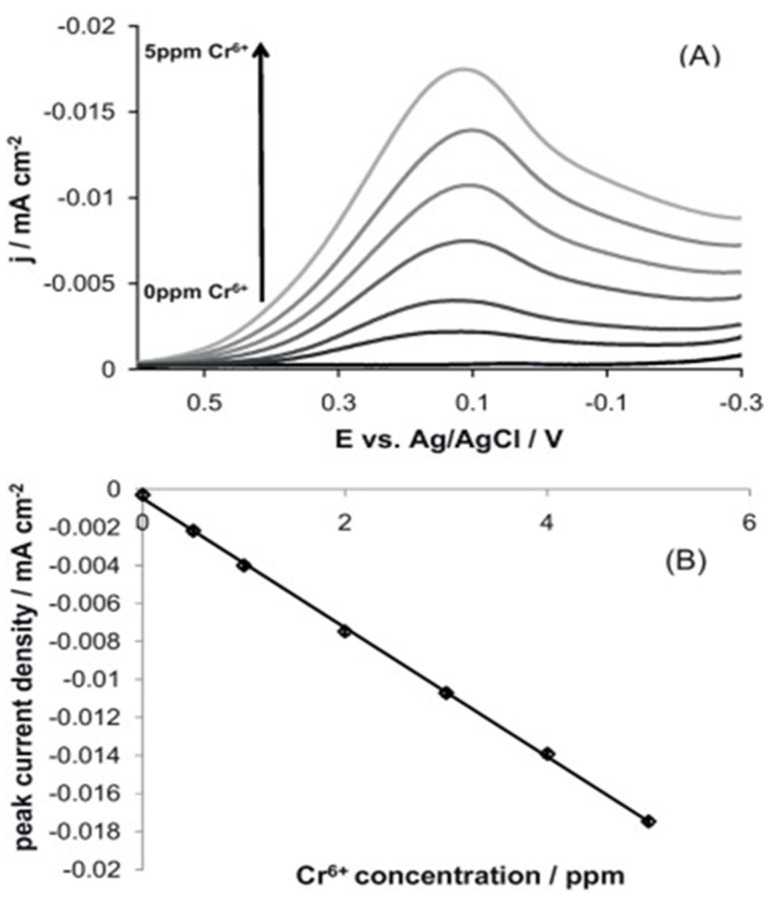
(**A**) Linear scan voltammograms of Cr^6+^ (concentration range between 500 ppb and 5 ppm) in 0.1 M HNO_3_ recorded on a BDD electrode at 50 mV s^−1^ and between 0.6 and −0.3 V vs. Ag/AgCl sat. T = 23 °C. (**B**) Chromium detection calibration curve: the peak current density reported in (**A**) was plotted as a function of Cr^6+^ concentration (slope: −3.41 × 10^−3^ ± 3 × 10^−5^, y-intercept: −4.8 × 10^−4^ ± 9 × 10^−5^, r^2^: 0.999) (reproduced with the permission of [78]).

**Figure 7 nanomaterials-12-00064-f007:**
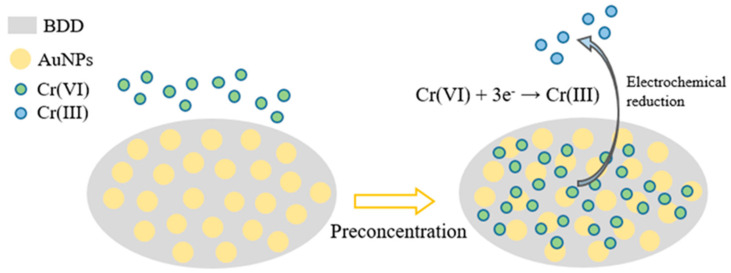
Working principle of Cr (VI) detection by gold nanoparticles (AuNPs)–boron-doped diamond (BDD) electrode (reproduced with the permission of [79]).

**Figure 8 nanomaterials-12-00064-f008:**
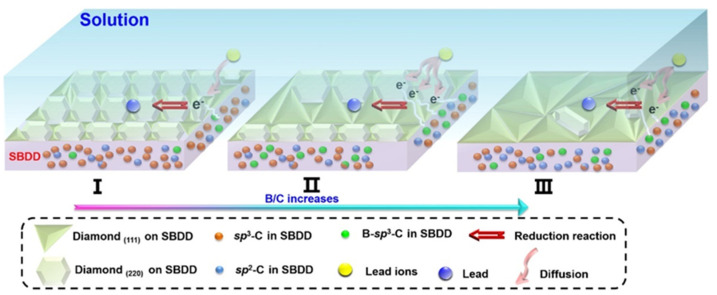
Schematic of the influence mechanism of B/C ratio on the detection performance of SBDD electrode for Pb^2+^ (reproduced with the permission of [90]).

**Figure 9 nanomaterials-12-00064-f009:**
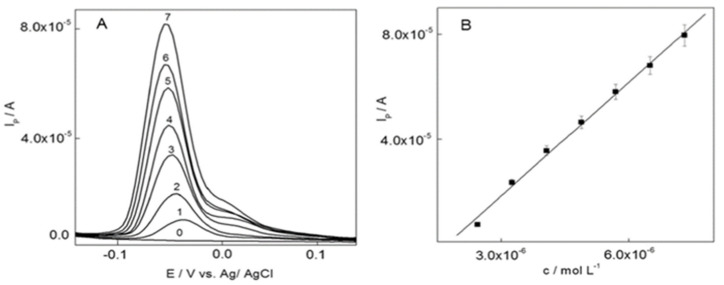
(**A**): Differential pulse anodic stripping voltammograms for 1 µL of 2.034 ×10^−2^ mol L^−1^ SbCl_3_ additions in 6 mol L^−1^ HClO_4_. Concentrations of Sb^3+^: (0) 0, (1) 2.44 × 10^−6^, (2) 3.25 × 10^−6^, (3) 4.07 ×10^−6^, (4) 4.88 × 10^−6^, (5) 5.69 × 10^−6^, (6) 6.50 × 10^−6^, and (7) 7.31 × 10^−6^ mol L^−1^ of Sb^3+^ on BDD electrode. Parameters: scan rate 25 mV s^−1^, modulation amplitude 40 mV, modulation time 50 s, deposition potential −1 V vs. Ag /AgCl and deposition time 240 s (**B**): Calibration curve for this experiment. Linearity was verified by F-test (reproduced with the permission of ref [93]).

**Figure 10 nanomaterials-12-00064-f010:**
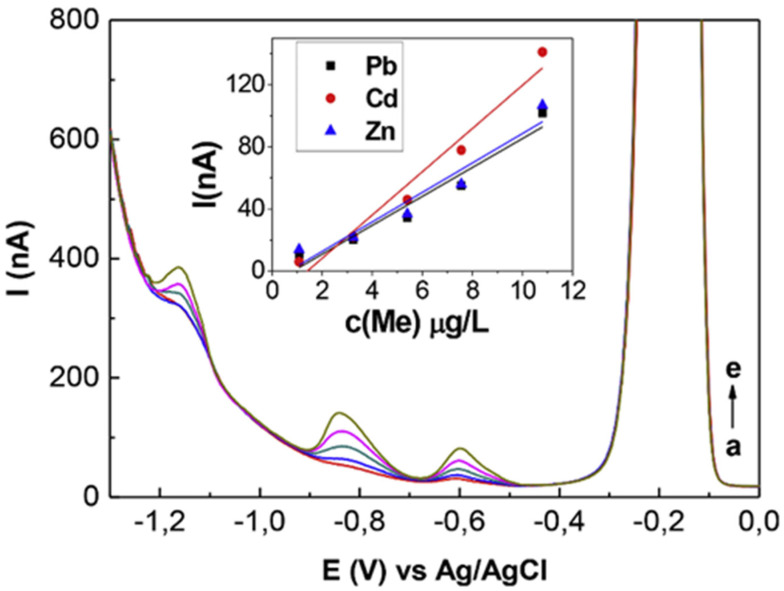
Square wave voltammograms for different additions of heavy metals with related calibration lines (electrode: CH_4_/H_2_ = 1%, C_B-C_ = 15,000 ppm, c (Bi) = 10 μg/L, (a) c (Me) = 1 μg/L, (b) c (Me) = 3 μg/L, (c) c (Me) = 5 μg/L, (d) c (Me) = 7 μg/L, (e) c (Me) = 10 μg/L) (reproduced with the permission of [97]).

**Figure 11 nanomaterials-12-00064-f011:**
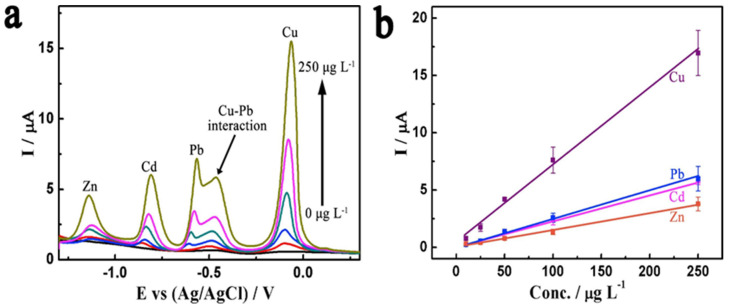
(**a**) Differential pulse anodic stripping voltammetry and (**b**) corresponding calibration plots for simultaneous analysis of Zn^2+^, Cd^2+^, Pb^2+^, and Cu^2+^ obtained on a D/G nanoplatelets film electrode. Error bar: n = 3 (reproduced with the permission of [103]).

**Figure 12 nanomaterials-12-00064-f012:**
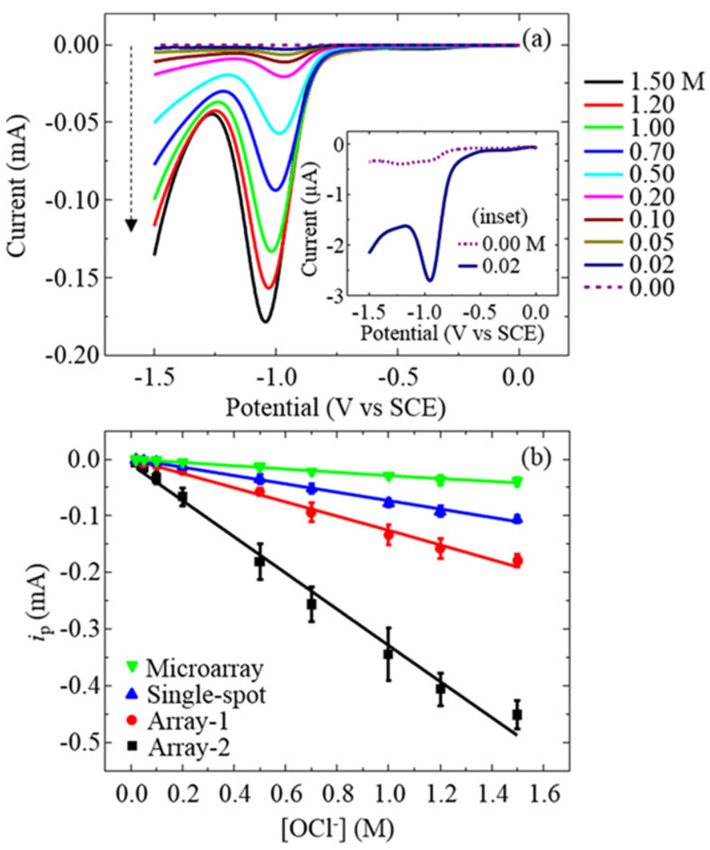
(**a**) Representative SWVs for the full concentration range (0.00−1.50 M OCl^−^) using the array-1 electrode. The insets show the lowest concentrations examined. The dashed arrow points toward increasing OCl^−^ concentrations. The SWV data is collected at 40 Hz with a perturbation amplitude of 0.05 V and data collection every 0.001 V (i.e., the effective scan rate is 0.04 V s^−1^). (**b**) Background-subtracted peak currents from SWV data for all electrodes as a function of [OCl^−^]. Error bars represent the sample standard deviation from n ≥ 3 measurements, and some error bars are contained within the symbols (reproduced with the permission of [111]).

**Figure 13 nanomaterials-12-00064-f013:**
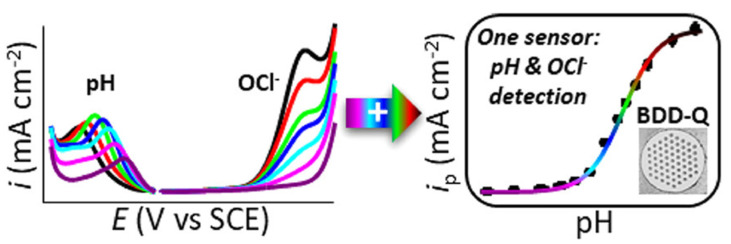
Schematic representation of sp^2^-BDD electrode-mediated detection of pH and OCl^−^ (reproduced with the permission of [112]).

**Figure 14 nanomaterials-12-00064-f014:**
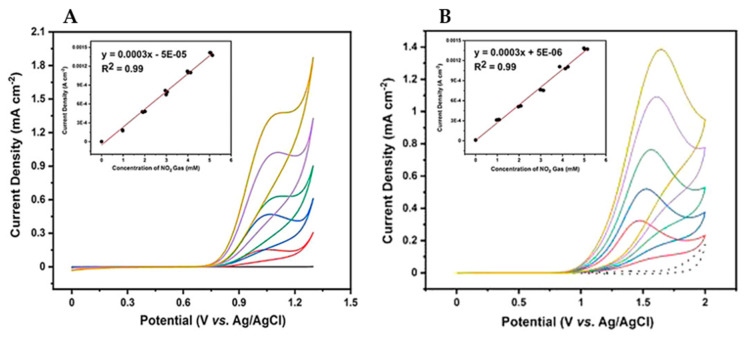
(**A**) Cyclic voltammograms in the concentration range of ~1 to 5 mM NO_2_ in a 0.1 M KClO_4_ solution at potentials of +1.1 V (vs. Ag/AgCl) and (**B**) +1.5 V (vs. Ag/AgCl) (insets: plots of current versus NaNO_2_ concentration) (reproduced with the permission of [115]).

**Table 1 nanomaterials-12-00064-t001:** Summary of diamond-based electrodes in the detection of metal ions.

Analyte	Electrode	Doping Concentration/Doping Atom	Method of Detection	Linear Range	Detection Limit (LOD)	Ref.
Ag^+^	Planar BDD/Disk BDD	10 ppm/Boron	DPASV	45.5 nM–2.3 µM and 9.1–682 nM	31 nM and 43 nM	[62]
Ag^+^	BDD	2000 ppm/Boron	DPASV	1–7 nM	0.2 nM	[63]
Ag^+^ and Cu^2+^	Diamond/Graphite film	n/a	ASV	0.91 pM–9 nM and 0.16 pM–15.7 nM, respectively	0.52 pM and 0.88 pM, respectively	[64]
Ag^+^ and Pb^2+^	BDD	n/a	ASV	0.75–0.025 mM and 0.72–0.05 mM, respectively	n/a	[65]
As^3+^	Ir-BDD	B/C = 1:100/Boron; implanted with 800 keV Ir^+^	Amperometry	0.1–100 µM	20 nM	[66]
As^3+^	Au-BDD	10^20^ cm^−3^	DPASV	0.133–534 pM	0.067 pM	[67]
As^3+^	AuNPs/BDD	n/a	SWASV	1.33–20 nM	13.35 nM	[68]
As^3+^	AuNPs- BDD	B/C = 10^4^ ppm/Boron	ASV	0–100 µM	64 nM	[69]
As^3+^	stable Ir-BDD	B/C = 0.1%/Boron	CV	0–100 µM	4.64 µM	[70]
Cd^2+^	BDD	B/C = 1%/Boron	ASV	0–0.1 mM	3. 94 nM	[71]
Cd^2+^	BDD	8000 ppm/Boron	SWASV	0.18–2.17 µM	8.9 nM	[72]
Cd^2+^	BDD	n/a	SWASV	18–445 pM	1.8 pM	[73]
Cd^2 +^ and Pb^2 +^	BDD	B/C = 2.4%/Boron	SWASV	0.05–4 µM and 0.05–8 µM, respectively	30.16 nM and 17.47 nM, respectively	[75]
Cd^2+^ and Pb^2+^	BDD	10000 ppm/Boron	DPASV	n/a	n/a	[76]
Cd^2+^ and Pb^2+^	D/CNWs	D/CNWs = 3%, 5%, 7% and 9%	DPASV	0.089–8.9 µM, and 0.048–4.83 µM, respectively	89 nM and 48 nM, respectively	[77]
Cr^6+^	BDD	B/C = 0.1%/Boron	LSV	0.2–96 nM	0.57 pM	[78]
Cr^6+^	AuNPs–BDD	n/a	SWCSV	0.193–19.3 µM	22.91 nM	[79]
Cu^2+^	BDD	2000 ppm/Boron	DPV	10 µM–100 mM	10 µM	[80]
Hg^2+^	BDD	n/a	DPV	10 nM–10 µM	68 nM	[81]
Hg^2+^	AuNPs–BDD	>10^20^ cm^−3^/Boron	SWASV	0.5–100 µM	5 µM	[82]
Hg^2+^	AuNPs–BDD	>10^20^ cm^−3^/Boron	EIS	1 pM–1 mM	n/a	[83]
Ni^2+^	BDD	~10^20^ cm^−3^/Boron	DPV	10–500 µM	26.1 µM	[84]
Ni^2+^ and Ni(OH)_2_ NPs	BDD	n/a	ASV	5–25 mM	5.73 µM	[85]
Ni(OH)_2_ NPs	BDD	n/a	ASV	10–25 mM	0.42 µM	[86]
Pb^2 +^	BDD	n/a	SWASV	9.65–145 nM	1.45 nM	[87]
Pb^2+^	BDD	B/C = 1000 ppm/Boron	SWASV	96–482 pM	19 pM	[88]
Pb^2+^	BDD	B/C = 2000 ppm/Boron	SWASV	4.83–48.3 nM	< 4.83 nM	[89]
Pb^2+^	SBDD	B/C = 0.05%, 0.1%, 0.2% and 0.4%/Boron	SWASV	15–362 nM	3.38 nM	[90]
Pb^2+^ and Cd^2+^	N-DNRs	n/a	SWASV	50 nM–1 µM and 10 nM–1.1 µM, respectively	50 and 10 nM, respectively	[91]
Pb^2+^ and Cu^2+^	BDD	8000 ppm/Boron	SWASV	30–180 nM (for both)	27 and 4 nM, respectively	[92]
Sb^3+^	BDD	1000 ppm or 10^20^ cm^−3^/Boron	DPASV	2.44–7.31 µM	108 nM	[93]
Zn^2+^	BDD	n/a	DPASV	0.5 nM–5 µM	0.47 nM	[94]
Pb^2+^, Cu^2+^, and Hg^2+^	BD-NCD	B/C = 0.3%/Boron	LSV	1–22.5 µM (for Pb^2+^ and Cu^2+^) and 1–10 µM (for Hg^2+^)	1.399, 0.102, and 0.666 µM, respectively	[95]
Ni^2+^, Cd^2+^, and Pb^2+^	PVC-BDD	7000–8000 ppm/Boron	DPASV	0–100 nM, 0–40 nM, and 0–150 nM, respectively	0.00424, 0.0221, and 0.25 nM, respectively	[96]
Zn^2+^, Cd^2+^, and Pb^2+^	Bismuth modified BDD	10000 ppm/Boron	SWASV	15–183 nM, 9–105 nM, and 5–58 nM, respectively	1.97, 0.57, and 0.51 nM, respectively	[97]
Ag^+^_,_ Cu^2+^, Pb^2+^, Cd^2+^, and Zn^2+^	NCD	n/a	DPASV	n/a	n/a	[98]
Zn^2+^, Cd^2+,^ Pb^2+^, and Cu^2+^	BDD	1300 ppm/Boron	DPASV	76–306 pM, 11–219 pM, 18.3–217 pM, and 47.2–315 pM, respectively	24, 3.16, 5.55, and 14.2 pM, respectively	[99]
Cd^2+^, Pb^2+^, Cu^2+^, and Hg^2+^	BDD	n/a	DPASV	0.088–0.88 nM, 0.048–0.48 nM, 0.157–1.57 nM, and 0.05–0.5 nM, respectively	30, 9.65, 1.57, and 3.49 pM, respectively	[100]
Pb^2+^, Cd^2+^, Zn^2+^, and Cu^2+^	BDD	7000–8000 ppm/Boron	ASV	n/a	n/a	[101]
Zn^2+^, Cd^2+^, Pb^2+^, and Cu^2+^	D/G nanoplatelets	n/a	DPASV	0.153–3.8 µM, 0.088–2.2 µM, 0.121–1.21 µM, and 0.157–3.93 µM, respectively	26.3, 4.13, 23.5, and 7.1 nM, respectively	[102]
Fe^3+^, Cu^2^+, Zn^2+^, Pb^2+^, and Cd^2+^	BDD	8000 ppm/Boron	SWASV	36–716 nM, 31–629 nM, 31–612 nM, 9.6–193 nM, and 17.56–351 nM, respectively	35.45, 22.34, 27, 8.4, and 14 nM, respectively	[103]

n/a = Not available.

**Table 2 nanomaterials-12-00064-t002:** Summary of diamond-based electrodes in the detection of anions.

Analyte	Electrode	Doping Concentration/Doping Atom	Method of Detection	Linear Range	Detection Limit (LOD)	Ref
N_3_^−^	BDD	1 × 10^19^ cm^−3^/Boron	LSV, DPV, and flow injection analysis	3.3 mM–0.30 µM	8 nM	[104]
H_2_O_2_	Pt-BDD	B/C = 1: 100/Boron and 5 × 10^14^ cm^−2^/Platinum	CV and flow injection analysis	0.1 to 10 µM	30 nM	[105]
H_2_O_2_	BDD/PB	10^19^–10^20^ cm^−3^/Boron	CV and flow injection analysis	n/a	n/a	[106]
I^−^	Diamond paste electrode	n/a	DPV	At pM–nM level	Subnanomolar level	[108]
I^−^/I_2_	BDD	0.1% wt/Boron	CV	0–1.2 mM/0–0.6 mM	20 µM/10 µM	[109]
F^−^ at [FeF_6_]^3−^/[CeF_6_]^2−^/[FeF_6_]	BDD	n/a	LSV and SWV	n/a	5 µM (LSV, [FeF_6_]^3−^), and 0.6 µM (SWV, [CeF_6_]^2−^)	[110]
OCl^−^	BDD	10^20^ atoms cm^−3^/Boron	SWV and LSV	0.02–1.5 M (by both)	n/a	[111]
OCl^−^	sp^2^-bonded carbon microspot-BDD	10^20^ atoms cm^−3^/Boron	SWV and LSV	58.31 µM–1.9 mM	58.31 µM	[112]
RNO_2_^−^	BDD	n/a	DPV	0.99–17 µM (absence of O_2_ at pH 4.0) and 0.99–11 µM (presence of O_2_ at pH 8.0)	0.41 µM (absence of O_2_ at pH 4.0) and 0.34 µM (presence of O_2_ at pH 8.0)	[113]
NO_2_^−^	BDD	7000–8000 ppm/Boron	SWV	4 µM–4 mM	20 µM	[114]
NO_2_^−^ and HONO	BDD	B/C = 1%	CV	1–5 mM (for both)	0.24 and 1.27 nM, respectively	[115]
NO_3_^−^	BDD	n/a	LSV	0–100 µM	1.5 µM	[116]
NO_3_^−^	BDD	B/C = 0.1%, 1%, 2%, and 3%	LSV	n/a	n/a	[117]
S_2_O_8_^2−^	BDD	500–8000 ppm/Boron	SWV	n/a	n/a	[118]
C_2_O_4_^2−^	ATAB-BDD	B/C = 10,000 ppm	Amperometry	0.8–100 µM	32 nM	[120]
S^2−^ and NO_2_^−^	BDD	B/C = 0.1%	CV, SWV and DPV	0.02–0.1 mM	n/a	[122]
NaNO_2_, CCl_3,_ COOH, and H_2_O_2_	Nafion/Mb/ND/CILE	n/a	CV	0.02–6.60 mM, 1.1–30 mM, and 0.3–19 mM, respectively	6.67, 370, and 100 µM, respectively	[124]
NaNO_2_, CCl_3,_ COOH, and KBrO_3_	Nafion/Hb/AuNPs/ND/CILE	n/a	CV	0.07–2.6 mM, 1–500 mM, and 0.35–12 mM, respectively	27, 330, and 3.3 µM, respectively	[125]

n/a = Not available.

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
