# Peer review of "Diamond-Based Electrodes for Detection of Metal Ions and Anions"

_nanomaterials, 2021, doi:10.3390/nano12010064_

Round 1

Reviewer 1 Report

This paper reports an original and interesting review dealing with the Diamond-Based Electrodes for Detection of Metal Ions and Anions. The ions have been analyzed according the existing literature. The results obtained by the authors and other authors highlight the good performance of BDD sensor. Relevant and sound results have been discussed. I recommend its publication with attention to the following minor comments:

I suggest rewrite all equations. Example: Ir (reduction form) à Ir (oxides form) ---------- (1)

Author Response

As suggested by the reviewer, we re-write the equations (1) and (2) and double check all other equations for readers clarity. Notably, except equations (1) and (2) all other equations are seems to be fine with respect to the original reports.

Reviewer 2 Report

the presented review represend a well done work of the authors, it is well organized, clearly presented, easy to read and does not impede the reader's flow. In addition it contains many practical informations accessible without extensive searching in the literature.

To date several reviews dealing with BDD material were published, but they are focussed to organic molecules, and not like this one to inorganic metal ions  and inorganic anions as also very important analytes.

these observations make presented review highly original and I am recommend it for publication in Nanomaterials journal. I strongly believe that it will strogly contribute to journal quality enhancement.

i have minor comments to enhace quality of presentation as follows

  1. line 214- were instead wee
  2. table 1 and table 2 are too extensive, they should be alingmented to one line with alphabetical order of analytes, and only the best and substantial data should be presented in one line for each analyte.
  3. please use for not assumed abreviation n / a instead NA

Author Response

  1. line 214- were instead wee

“Author Response”

As per reviewer’s advice those typo errors were rectified in the revised version.

  1. table 1 and table 2 are too extensive, they should be alingmented to one line with alphabetical order of analytes, and only the best and substantial data should be presented in one line for each analyte.

“Author Response”

As per reviewer’s suggestion, Tables 1 and 2 are aligned in one line with respect to discussions of each section and the essential data are provided in the table for reader’s clarity. Alignment with respect to alphabetical order of analytes may affect the readers’ clarity due to diverse reference numbers. So, at this stage it is not possible to modify the Tables further.

  1. please use for not assumed abreviation n / a instead NA

“Author Response”

As suggested by the reviewer, the abbreviation n / a is included instead of NA.